# Implications of tissue specific STING protein flux and abundance on inflammation and the development of targeted therapeutics

**Thomas E. Angel[1], Zhuo Chen[1], Ahmed Moghieb[1], Sze-Ling Ng[2], Allison M. Beal[2], Carol Capriotti[2], Leonard Azzarano[1], Debra Comroe[1], Michael Adam[3], Patrick Moore[2], Bao Hoang[1], Kelly Blough[1], Joanne Kuziw[1], Joshi M. Ramanjulu[2], G. Scott Pesiridis** [4]*

**1** In vitro/In vivo Translation, Research and Development, GlaxoSmithKline, Collegeville, Pennsylvania, United States of America, **2** Respiratory and Immunology Research Unit, Research and Development, GlaxoSmithKline, Collegeville, Pennsylvania, United States of America, **3** Oncology Extracellular Targeted Cancer Therapeutics, Research and Development, GlaxoSmithKline, Collegeville, Pennsylvania, United States of America, **4** Discovery Project Leadership Team, Research and Development, GlaxoSmithKline, Collegeville, Pennsylvania, United States of America

* g.s.pesiridis@gsk.com

## Abstract

Drugs targeting the ER-resident innate immune receptor Stimulator of Interferon Genes (STING) are in development for treatments of cancer and inflammatory diseases. Accurate determination of STING receptor levels in normal and disease tissue is an essential component of modeling pharmacology and drug-target disposition. Using metabolic labeling with deuterium oxide paired with high resolution mass spectrometry, we report the protein fractional synthesis rates and turnover of STING in wild-type (C57BL/6) and inflamed mice carrying the Trex1 D18N mutation (Trex1$^{D18N}$) as a STING-dependent model of human Acardi-Goutiéres syndrome. Remarkably, STING protein half-life is tissue specific with the shortest half-life of 4 days in colon and lymph node and longest half-life of 24 days in skeletal muscle. Despite the relative increase in STING protein abundance in the inflamed Trex1$^{D18N}$ mouse, the overall kinetics of protein degradation and resynthesis was similar between Trex1$^{D18N}$ and WT mice. The extent of tissue specific interferon stimulated gene transcription, a hallmark of SLE linked pathophysiology, correlates with the extend of increased STING levels in Trex1$^{D18N}$ tissues and appears inversely proportional to the turnover rate of STING. Understanding STING's fractional protein synthesis rate and half-life provides a valuable component of quantitative modeling of drug pharmacology, dose frequency and targeting tissues of STING directed therapies.

## Introduction

To-date there are multiple STING agonist assets in clinical development as a monotherapy or in combination therapy across > 25 human trials in oncology. However, the field has stagnated with less than impressive clinical outcomes. One hypothesis for sub-par efficacy is that the field lacks a quantitative understanding of STING's protein turnover that dictates dose frequency and receptor activation. Importantly, activation of STING initiates a cascade of events

**Data availability statement:** All relevant data are within the manuscript and its Supporting information files.

**Funding:** The author(s) received no specific funding for this work.

**Competing interests:** All authors listed are current or former employees of GSK who have owned shares in the past or currently hold shares in GSK plc. This does not alter our adherence to PLOS ONE policies on sharing data and materials.

that lead to its degradation as a negative feedback mechanism to desensitize cells during the STING response [1]. This observation raises the question as to how newly synthesized forms of apo-STING may replenish receptor levels following pharmacological activation or whether STING levels are rapidly turning over in an inflamed tissue environment.

STING is an integral membrane protein that resides on endoplasmic reticulum as a dimer [2]. It is expressed in many cell types (macrophages, DCs, T cells, endothelial, fibroblast, and epithelial), tissues, and organs, with the highest levels reported in organs associated with the innate immune response and the lowest levels in liver and skin [3]. STING is downstream of the cytosolic DNA sensor cyclic guanine adenine synthetase (cGAS) which catalyzes the formation of the STING-activating ligand cGAMP (cyclic guanine adenine monophosphate) and mediates the transcription of interferons and proinflammatory genes [4,5]. Activation of STING results in its translocation from the endoplasmic reticulum (ER) to perinuclear endosomes where STING forms a scaffold that facilitates interactions between activating kinases, TANK-binding kinase 1 (TBK1) and IkB kinase (IKK), that phosphorylate the transcription factors interferon regulatory factor 3 (IRF3) and nuclear factor kappa-light-chain-enhancer of activated B cells (NF-κB).

Dysregulation of the cGAS-STING pathway plays a pathophysiological role in spectrum of human inflammatory diseases, neurodegenerative disease, liver disease, infectious diseases and cancer [5]. The strongest data and genetic evidence support the development of cGAS-STING pathway inhibitors in systemic lupus erythematosus (SLE) and related auto-inflammatory syndromes. Multiple loss-of-function mutations in the DNA nuclease TREX1 that negatively regulate the DNA stimuli of cGAS-STING results in Aicardi-Goutières syndrome and familial chilblain lupus. For instance, the D18N mutation in the TREX1 gene impairs exonuclease function and cannot effectively degrade double-stranded DNA [6,7]. Impaired degradation of double-stranded DNA triggers cGAS and STING, activating a downstream interferon-mediated immune response and systemic lupus-like pathology. The D18N mutation of Trex1 in mice recapitulates many features of SLE pathology and familial chilblain lupus, including systemic inflammation, lymphoid hyperplasia, anemia, vasculitis, and kidney disease [8–10]. Deletion of cGAS or STING rescues the inflammatory phenotype of Trex1$^{D18N}$ mice as well as the more severe inflammatory phenotypes resulting from the complete knock out of Trex1 [9–11]. As the mechanistic understanding of cGAS-STING pathway has rapidly advanced in recent years, increasing efforts of anti-inflammatory and anti-cancer therapeutics development are focusing on modulating STING protein [12–14]. For better therapeutics and dosing design, a clearer understanding of tissue-specific half-life of STING is of high importance.

Protein identification and quantitation are not always sufficient to fully understand the complexities in most if not all biological systems. Proteins in tissue and organs are temporally and chemically dynamic, abundance based static data from traditional protein quantitation ignores the kinetic aspect. There are many examples of protein biomarkers that have been identified, yet few have translated to clinical utility [15–17]. Protein abundance is an integrated measure of past influences leading to a biological steady state. Lack of successful translation for putative biomarkers to clinical utility is partially due to large variation in biomarker abundance due to uncontrolled or un-isolated factors driving changes in biomarker abundance resulting in heterogeneity across conditions being compared. Application of stable isotope labelling protocols allow for temporal isolation of biological processes such as protein synthesis and degradation. Historically, measurement of stable isotope label incorporation has led to critically important discoveries in biological sciences, resulting in increased understanding into fundamental biological processes [18,19]. Past success with stable isotope labelling leading to understanding of protein dynamics (protein synthesis and degradation) enabling insight into biology associated with disease represent the foundation

of future efforts which represent the next generation of protein biomarker characterization and discovery [20–24].

Protein synthesis rates can be measured *in vivo* by introducing a stable isotopic label (e.g., $^2$H, $^{13}$C, $^{15}$N) and monitoring the rate of incorporation over time. Deuterium oxide can be applied and used as a universal stable isotope tracer with the advantages that it is non-toxic at low exposure levels (< 20% mole percent excess (MPE)) [25], can be taken orally, rapidly mixes through out the entire body, and as it is not distinguished with regular water it is readily incorporated into metabolic precursor pools in predictable and reproducible ways [26,27]. The combination of metabolic stable isotope labeling and mass spectrometry-based protein analysis enables both untargeted and targeted analysis of protein dynamics (fractional synthesis rate or half-life) of biological systems at the intact organismal level [23,28–33].

Metabolic labelling with stable isotopes combined with mass isotopomer distribution analysis has been applied to the analysis of a range of molecules including lipids [34,35], metabolites [36], and proteins [21,23,37–41]. Allometric analysis, quantification of differential sizes and rates of organ or tissue growth, quantifying protein fractional synthesis following administration of stable isotope label demonstrated important differences in protein dynamics between organs [42] and between different preclinical animal models and humans [39].

In this study, we report the fractional synthesis rate and corresponding half-lives of STING in eight different tissues (lymph node, thymus, spleen, heart, skeletal muscle, colon, and salivary gland) reflecting a range of biological compartments in wild type (C57BL/6) and chronically inflamed Trex1$^{D18N}$ mice following metabolic labelling with deuterium oxide.

## Materials and methods

### Chemicals and reagents

Recombinant Mouse STING protein (aa 160-371) was purchased from LifeSpan BioSciences (WA, USA). Pierce™ IP lysis buffer and SuperBlock™ blocking buffer were purchased from Thermo Scientific (MA, USA). LC/MS solvents including water, acetonitrile, methanol, and formic acid were purchased from Fisher Scientific (MA, USA).

### diABZI dependent degradation of STING *in vitro* and *in vivo*

THP-1 cells grown in RPMI 1640 (Gibco) + 10% FBS treated with 1 μM diABZI STING agonist [14] for 0, 30, 60, 120, 360 and 1440 minutes. Cells lysates were prepared for WES analysis (ProteinSimple) as previously described [14] using anti-STING (Cell Signaling), anti-phospho-STING (Cell Signaling), and anti-vinculin (Sigma) as the loading control.

STING protein levels were measured by immunohistochemical staining of STING in BALB/c mice containing a colon carcinoma cell, CT-26, derived tumor. Six to eight-week-old female BALB/c mice were ordered from Envigo. CT-26 cells (ATCC: CRL-2638) were thawed, plated in RPMI with 10% FBS and 1% PS, and sub-cultured 3 times over 6 days prior to use. Cells were suspended in RPMI at 5 x 10$^5$ cells/ml and inoculated subcutaneously with 0.1 mL CT-26 cells in RPMI on the right hind flank on day -14. On day 0, diABZI [14] was freshly prepared in 40% PEG400/saline vehicle at a concentration of 1 mg/ml and dosed intratumorally with 50 μL dosing solution (50 μg/mouse). STING protein levels in Trex1$^{D18N}$ heart was measured by immunohistochemical staining of STING harvested from 2-month old Trex1$^{D18N}$ mice.

Tumors and hearts were freshly prepared using the same process. Excised tissues were trimmed and formalin fixed paraffin embedded for immunohistochemistry. 3.5 micron sections were mounted on a Ventana Discovery Ultra system (Tucson, AZ), incubated for 20 mins at 60°C, followed by three 8 minute cycles of deparaffinization. Antigen retrieval was

performed using Tris-based (EDTA) buffer solution, CC1 (Roche). Tissue sections were incubated with either rabbit anti-STING, clone D2P2F (Cell Signaling, 1:100 dilution) or rabbit IgG isotype control (Roche, predilute, 790-4795) for 32 mins at 37°C. Secondary antibody detection was achieved by incubation with anti-Rb HQ for 16 minutes at 37°C prior to incubation with anti HQ-HRP (Roche) for 16 minutes. Tissue sections were visualized using ChromaMap DAB detection kit (Roche) and counterstaining with hematoxylin II (Roche). Whole slide digital images were captured using the Nanozoomer slide scanner (Hamamatsu, Bridgewater, NJ).

## Animal model

**Generation of Trex1$^{D18N}$ mice.** Trex1 D18N mice were generated by homologous recombination using a targeting vector with a single D18N mutation in exon 5 (GenoWay) and have been designated the name C57BL/6NCrl-*Trex1$^{tm1.1(D18N)Geno}$*/Gsk, herein Trex1$^{D18N}$ (S1 Fig). A Neomycin cassette flanked by loxP sites was included for positive selection. The targeting vector was constructed using C57BL/6 mouse genomic DNA and electroporated into isogenic ES cells. Homozygous Trex1$^{D18N}$ mice were generated using standard homologous recombination techniques and breeding.

**Dosing protocol.** Based on gender and genotype, mice were categorized into four cohorts: 1) female, wild type (C57BL/6); 2) female, Trex1$^{D18N}$ homozygous; 3) male, wild type (C57BL/6); 4) male, Trex1$^{D18N}$ homozygous mice. In each cohort, mice were divided into one control group and five dosing groups (n = 5 per group). Mice from five dosing groups were dosed intraperitoneal (IP), with deuterium oxide (0.9% w/v NaCl, 99.9% 2H2O - Sigma cat# 151882). After IP bolus, mice were provided free access to food and enriched drinking water (8% molar enrichment 2H2O) for the remainder of the experiment. Mice labeled for 1, 3, 7, 15, and 21 days were euthanized with isoflurane and exsanguinated via cardiac puncture. Blood samples were collected into EDTA tubes, centrifuged, and stored at -20°C until analysis. Tissues were collected and stored at −20°C until analysis. All studies were conducted in accordance with the GSK Policy on the Care, Welfare and Treatment of Laboratory Animals and were reviewed by the Institutional Animal Care and Use Committee at GSK.

## FSR determination

**Deuterium oxide enrichment.** Plasma was stored at −20°C until deuterium enrichment could be measured. MPE was measured in the samples against an accompanying standard curve using a cavity ring-down water isotope analyzer (Los Gatos Research, CA, USA) according to previous published method [43].

**Tissue homogenization.** Mouse tissue (~30 mg) was placed in Precellys® homogenization tubes (Cayman Chemical, MI, USA) containing 750 µL of Pierce™ IP Lysis Buffer. Homogenate samples on Bead Mill 24 (Thermo Scientific, MA, USA) at 6 m/sec for 30 seconds and centrifuge samples at 15,000 g for 5 minutes at room temperature. 200 µL of supernatant from each sample was then used for immunocapture.

**Immunocapture and on-plate trypsin digestion.** A MaxiSorp plate (Thermo Scientific, MA, USA) was coated with 100 µL of 10 µg/mL anti-mouse STING antibody (cat.# MABF213, Millipore Sigma, MO, USA) on each well. The plate was incubated with constant shaking overnight at 4 °C, followed by blocking with SuperBlock™ T20 (TBS) blocking buffer for 2 hours at 37 °C. Mouse tissue homogenate was then loaded onto the plate and incubated for 2 hours at 37 °C. Immunocapture, 100 µL of 10 µg/mL trypsin in 100 mM sodium bicarbonate buffer (pH 8.5), was used for on-plate digestion. After digestion for overnight at 37 °C, 5 µL 10% formic acid in water was added to terminate the digestion and the prepared sample was ready for LC-MS analysis.

**LC-MS condition for immonium ion isotopologue distribution analysis.** Prepared samples were analyzed on Q Exactive Plus mass spectrometer (Thermo Fisher Scientific, MA, USA). Tryptic peptides were separated using EvoSep One system (EvoSep, Denmark) coupled to the Easy-Spray™ source (Thermo Fisher Scientific, MA, USA). Mobile phase A was 0.1% formic acid in water and mobile phase B was acetonitrile. Samples were loaded onto EvoSep C18 tip and analyzed using the 5.6 min gradient [44]. Parallel-reaction monitoring (PRM) mass spectrometric analysis on Q Exactive Plus was performed using 2.0 kV in the ion source, 140,000 resolution setting with a scan range of 50-1250 *m/z*, 1E6 AGC target, a maximum injection time of 100 msec, and a lock mass of *m/z* 445.12002. MS2 scans were triggered using a targeted inclusion list for +2 charge state for STING derived peptides DMLPQQNIDR (2-3.5 minutes), LILPGLQAR (3.5-4 minutes), TLEEILEDVPESR (4-5 minutes) and the fragmentation scan used an isolation window of 1.0 *m/z*, HCD fragmentation with an energy of 25% or 100% NCE and fixed first mass of 50.0 m/z ensuring observation of immonium ions. Profile spectra were collected facilitating immonium ion isotopologue analysis.

## Data processing

Peptides were identified from MS2 spectra using the Mascot search engine [45]. Trypsin was designated as the digestion enzyme and up to two missed cleavages were allowed. Identification database was uniprot validated database, restricted to *Mus musculus* protein entries. HCD fragmentation was specified, and peptide MS and MS/MS mass tolerances were set to 20 ppm.

Immonium ion quantitation of three STING derived DMLPQQNIDR, LILPGLQAR, and TLEEILEDVPESR was performed with Skyline v4.2. The exported signal intensity for immonium ion isotopologues were used to calculate fractional synthesis as described below: Immonium ion isotopologue distributions were modeled using combinatorial probability calculations. Briefly, natural abundances given by the National Institute of Standards and Technology (NIST) were used for all elements. The number of hydrogen to deuterium substitutions on each amino acid following metabolic labeling has been previously determined [46] and reported values were used in our analysis. Deuterium isotope labelling was represented as element X, abundances were altered without affecting the unlabeled hydrogen abundances. Normalized distribution of the M1 isotopologue peaks (each containing an extra neutron) of carbon and hydrogen were calculated for 0-5% MPE (Equations 1 and 2). These values for carbon and hydrogen isotopologue abundances were fit using 3rd order polynomials which have previously been shown to be sufficient for fitting changes in distribution of mass isotopomers for the MPE range in the current study [40]. We used the derived polynomial coefficients to calculate maximum normalized immonium ion abundances at the deuterium MPE measured for each sample.

$$\left[ M1\ ^{13}C \right] = \left( \left[ ^{12}C \right]^{\#C-1} * \left[ ^{13}C \right] \right) * \#C * \left[ ^{1}H \right]^{\#H} * \left( \left[ ^{1}H \right] - MPE \right)^{\#X} * \left[ ^{14}N \right]^{\#N} * \left[ ^{16}O \right]^{\#O} \quad (1)$$

$$\left[ M1\ ^{2}H \right] = \left[ ^{12}C \right]^{\#C} * \left( \left[ ^{1}H \right]^{\#H-1} * \left[ ^{2}H \right] \right) * \#H * \left( \left[ ^{1}H \right] - MPE \right)^{\#X} * \left[ ^{14}N \right]^{\#N} * \left[ ^{16}O \right]^{\#O} + \left[ ^{12}C \right]^{\#C} *$$
$$\left[ ^{1}H \right]^{\#H} * \left( \left( \left[ ^{1}H \right] - MPE \right)^{\#X-1} * \left( \left[ ^{2}H \right] + MPE \right) \right) * \#X * \left[ ^{14}N \right]^{\#N} * \left[ ^{16}O \right]^{\#O} \quad (2)$$

$$\text{Normalized 2H abundance} = 2H / \left( 2H + 13C \right) \quad (3)$$

is calculated from modeled or measured peak heights or areas.

$$f = \frac{\text{Normalized Measured2}_H \text{ abundance}}{\text{Normalized Theoretical2}_H \text{ abundance}} \text{ , (normalized, unitless)} \quad (4)$$

Fractional synthesis rate (FSR) is calculated using the exponential rise to plateau equation $f = 1-e^{-kt}$, where the rate constant k is synonymous with FSR, f is the measured fractional synthesis over the labeling interval t, for normalized $^2$H abundance (Equations 3 and 4).

**LC-MS condition for MRM analysis.** For multiple reaction monitoring (MRM) measurement, an ACQUITY UPLC System (Waters Corporation, MA, USA) was used for conventional flow UPLC analysis. 20 μL of prepared sample was injected onto an ACQUITY UPLC BEH C18 Column (300 Å, 1.7 μm, 2.1 mm x 50mm, Waters Corporation, MA, USA) at a temperature of 60 °C. Mobile phase A was 0.1% formic acid in water and mobile phase B was acetonitrile. Separation was executed at a flow rate of 0.6 mL/min with gradient as follows: 0-1.60 min 5-50% B, 1.61-1.99 min 95% B, 2.00-2.50 min 5% B.

Mass spectrometry analysis was performed using a Waters TQ-S in positive ESI mode. All data was acquired with Waters Masslynx Version 4.1 SCN905. Source temperature and capillary voltage were set to 150 °C and 2.0 kV, respectively. Cone gas flow was 150 L/hr. Desolvation temperature was set to 650 °C at a gas flow of 800 L/hr. Dwell time was set to 0.06 sec. MRM transitions and parameters were optimized as below: For LILPGLQAR, the precursor $[M+2H]^{2+}$ to $y6$ product ion transition ($m/z$ 490.82 to 641.37) was used with collision energy (CE) of 17 and cone voltage (CV) of 40; for DMLPQQNIDR, the precursor $[M+2H]^{2+}$ to $y7$ product ion transition ($m/z$ 615.30 to 870.44) was used with CE of 19 and CV of 20; for TLEEILEDVPESR, the precursor $[M+2H]^{2+}$ to $y8$ product ion transition ($m/z$ 765.39 to 944.47) was used with CE of 27 and CV of 20.

**qPCR analysis.** Tissues were harvested and snap frozen in liquid nitrogen prior to −80C storage. RNA was isolated from tissues by homogenization with Qiagen Tissue Lyser and RNeasy Extraction Kit. Synthesis of cDNA was performed using Applied Biosystems High Capacity cDNA Reverse Transcription Kit. QPCR was conducted with Applied Biosystems Master Mix and Life Technologies primer probes in an Applied BioSystems 7900 HT Fast Real-Time PCR System. Data was analyzed by DDCT method and 18S housekeeping gene was used for normalization. Fold change was calculated in reference to the average control group response; $2^{\Delta\Delta Ct}$ where $\Delta\Delta Ct = \overline{Ct_{WT}} - \left(\overline{Ct_x} - \overline{Ct_{18S}}\right)$, $\overline{Ct_x}$ is the average response from technical replicates of IFIT3, CXCL10, OAS1, and TNFα qPCR measurements and $\overline{Ct_{18S}}$ is the average response from technical replicates of the 18s ribosomal subunit. Statistical analysis using an unpaired t-test compared the mean of WT versus the mean of Trex1$^{D18N}$ fold change responses for each gene and reported a statistical significance represented as p-value intervals.

## Results

Previous studies demonstrated that STING protein levels, stability and function are regulated by complex mechanisms of ubiquitination that impact STING protein turnover and its ability to facilitate innate immune signaling [47–54]. It is known that activation of STING, either under innate immune stimulatory conditions or by agonist dependent activation, leads to endolysosomal ER-to-Golgi (ERGIC) trafficking and nearly complete endolysosomal degradation of STING (Fig 1A) [1]. Similarly, activation of STING *in vivo* by direct intratumoral injection with a potent STING agonist (diABZI) can cause complete loss of STING in a mouse CT-26 tumor microenvironment (Fig 1B). Initial estimates of STING protein half-life report a relatively long half-life of > 500 hours in primary human monocytes, NK-cells, and B-cells and is likely limited by a lack of unstimulated depletion over the course of the observation [55]. However, it is currently unknown how long it takes for STING levels to recover to a homeostatic state. This leaves an important unanswered question; if STING is degraded under activating conditions like

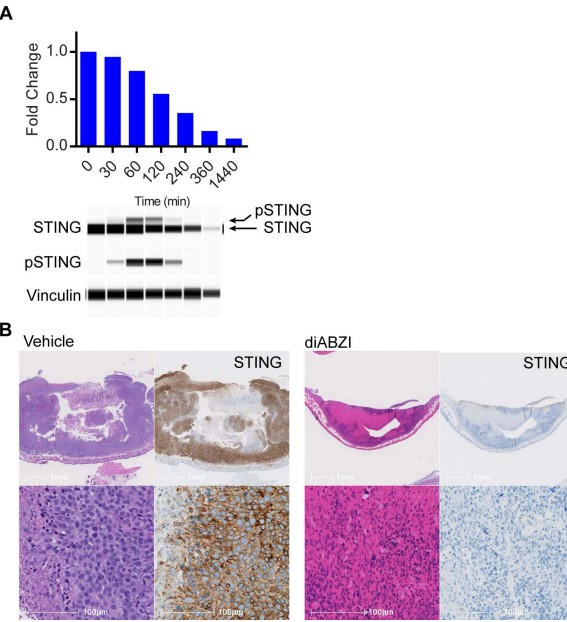

**Fig 1. STING protein clearance following STING agonist treatment.** A) STING protein levels were measured by WES ProteinSimple in cell extracts from THP-1 cells treated with 1 μM STING diABZI agonist. Panels are digitized capillary electrophoresis visualization of quantified STING, phospho-STING (pSTING) and loading control vinculin. B) STING immunohistochemistry in 100 mm³ CT-26 tumors treated with an intratumoral injection of 50 μg of diA-BZI STING agonist or vehicle (40% PEG400 in Saline). Tumors were harvested 24 hours after treatment and prepared for H&E stain or immunohistochemistry of total STING protein.

systemic inflammation or agonist activation, then what is the dynamic flux of STING in cells and tissues that drive disease pathophysiology and how does this impact drug pharmacology?

Aicardi-Goutières syndrome is a monogenic disease caused by mutations in the deoxyribonuclease TREX1 leading to chronic activation of the cGAS-STING pathway. To evaluate the fractional resynthesis rates of STING in a disease relevant model of inflammation, we generated a transgenic mouse model of AGS by mutation of the mouse Trex1 gene resulting in the D18N point mutation into C57BL6 mice to generate C57BL/6NCrl-*Trex1*$^{tm1.1(D18N)Geno}$/Gsk, herein Trex1$^{D18N}$ (S1 Fig). These animals recapitulate hallmark inflammatory phenotypes of AGS and are similar to previously published murine models carrying the Trex1$^{D18N}$ mutation [8]. Importantly, animals carrying homozygous D18N mutations succumb to an inflammatory phenotype over the course of 12–30 weeks with elevated blood levels of systemic cytokines (IP-10, MCP-1 and TNFα), increased levels of total IgG including disease-relevant autoantibodies like anti-dsDNA and anti-ssDNA (S1 Fig). Histopathological characterization demonstrates multiorgan inflammation in the heart, salivary gland, lung, kidney, and spleen (S1 Fig). Furthermore, knock-out of the mouse STING gene in the C57BL/6NCrl-*Trex1*$^{tm1.1(D18N)Geno}$/Gsk transgenic mouse rescued survival, tissue inflammation and spontaneous secretion of inflammatory cytokines [11]. Taken together, our Trex1$^{D18N}$ murine model recapitulates key hallmarks of inflammation and reflects immune phenotypes dependent on STING in distinct tissue environments that we aim to measure its fractional synthesis rates.

To understand the relationship between STING protein levels and fractional resynthesis rate in the Trex1$^{D18N}$ model, we first determined the relative abundance of STING using a multiple reaction monitoring (MRM) mass spectrometry-based detection of the top three tryptic peptides in STING (S2 Fig and Fig 2A). The relative STING abundance was variable between

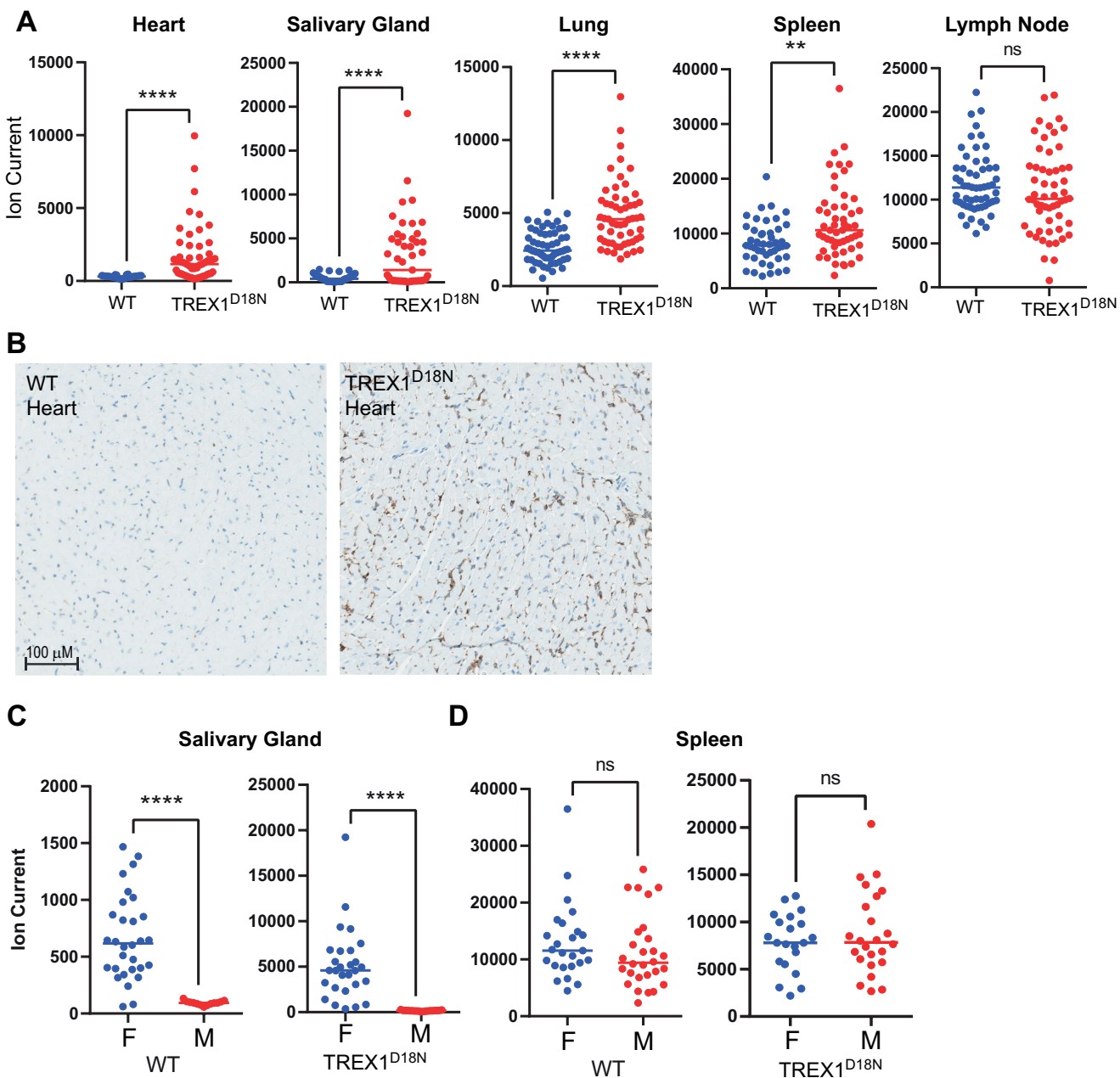

**Fig 2. STING protein levels *in vivo*.** A) STING protein levels measured by MRM mass spectrometry in mouse heart, salivary gland, lung, spleen, and lymph node from WT (blue) and TREX1[D18N] (red) mice. B) Immunohistochemistry staining of mouse STING in WT and TREX1[D18N] hearts. Scale bar reflects 100 μM. STING protein levels in female (F) and male (M) (C) salivary gland and (D) spleen. *ns,* not significant; ** and **** reflect t-test *p-values* of 0.001 and 0.0001, respectively.

individual animals and between different tissue types (Fig 2A). STING levels were higher in tissues from Trex1[D18N] mice with the greatest difference observed in heart, salivary gland, and lung (Fig 2A). There was a subtle, yet significant difference in spleen but differences in lymph node were not significant (Fig 2A). Immunohistochemical staining of STING in heart provided clear visual confirmation for the observed increase in STING protein levels (Fig 2B). The rank order of STING abundance was highest in spleen, lymph node, and lung followed

by heart and salivary gland. The salivary gland showed a higher level of STING protein when comparing female to male animals, however this gender specific trend was not observed in other tissues (Fig 2C).

To further quantify STING protein half-life we measured its FSR in eight tissues: lymph node, spleen, thymus, salivary gland, lung, heart, skeletal muscle, and colon. Metabolic labeling was achieved by administration of an initial bolus of deuterium oxide ($D_2O$) followed by introduction of 8% $D_2O$ in drinking water resulting in labeling of non-essential amino acids during *de novo* synthesis and the subsequently synthesized proteins. Tissues were harvested after 1, 3, 7, 14, and 21 days of $D_2O$ exposure and immunocapture enrichment of STING was performed before proteolysis and LC-MS quantification of STING protein levels (S2 Fig). Deuterium incorporation and enrichment into STING was quantified monitoring enrichment in proline amino acid side chain in the tryptic peptide LILPGLQAR (S2 Fig) [56].

There was a consistent increase in the deuterium signal abundance over time in the trypsin digested STING peptide LILPGLQAR signal abundance over time reflecting a time dependent accumulation of newly synthesized STING protein in all tissues examined (Fig 3A). The difference in FSR of STING was striking in that STING protein half-life appears unique to each tissue type (Fig 3B–J). The half-life of STING in the thymus, lymph node, and colon exhibited the shortest average half-life of ~4 days and is nearly 2-fold shorter than the half-life of STING protein in spleen, salivary gland, and lung with average half-life of approximately 7-8 days (Table 1, Figs 3B–K). Remarkably, the observed half-life of STING in heart and skeletal muscle was the longest at approximately 14 and 23 days, respectively. Despite gender specific differences in total abundance of STING (Fig 2D), there was no significant difference in FSR and corresponding STING protein half-life between male and female animals (Table 1). Taken together, this data demonstrates that STING protein exhibits a range of distinct protein turnover rates that are a unique property of the tissue environment and vary in range between 4 and 24 days.

STING half-life did not differ dramatically between WT and Trex1[D18N] animals except for the heart where the FSR was approximately 20% faster in Trex1[D18N] compared to WT animals (Fig 3I). This difference amounts to a small shift in the STING's half-life from 15 to 12 days (Table 1). Interestingly, STING-dependent inflammation in the heart is described as the key phenotypic driver of the survival phenotype in Trex1[−/−] and Trex1[D18N] mice [8,57]. This demonstrates the sensitivity of the FSR analysis for measuring perturbations in biological state. This data illustrates that the global FSR of STING protein is not dramatically impacted by inflammatory condition reflected in the Trex1[D18N] mice at 2 months of age with a significant trend toward a higher FSR in the inflamed heart.

To further correlate STING's tissue specific FSR and inflammation, interferon stimulated genes and TNFα were measured in heart, salivary gland, lung, and spleen. There was a statistically significant increase in IFIT3, CXCL10, OAS1, and TNFα measured in the Trex1[D18N] tissues compared to wild-type controls (Figs 4 and S3). The rank order of highest to lowest increase in IFIT3, OAS1, CXCL10, and TNFα followed a similar trend as the highest increase in STING protein levels observed in heart followed by salivary gland, lung, and spleen (Fig 2A). This trend follows a similar trend as the rank order of tissues with the highest increase in STING abundance when comparing Trex1[D81N] to WT controls (Fig 2A). Interestingly, the increase in gene expression appears inversely proportional with STING's tissue specific turnover rate. The heart demonstrates the slowest STING protein turnover among the set of tissues tested for gene expression, yet it demonstrates the highest levels of IFIT3, OAS1, CXCL10 and TNFα. It is also interesting to note that the spleen showed the highest overall abundance of STING with the fastest STING protein turnover yet the smallest increase in IFIT3, OAS1, CXCL10 and TNFα gene transcription.

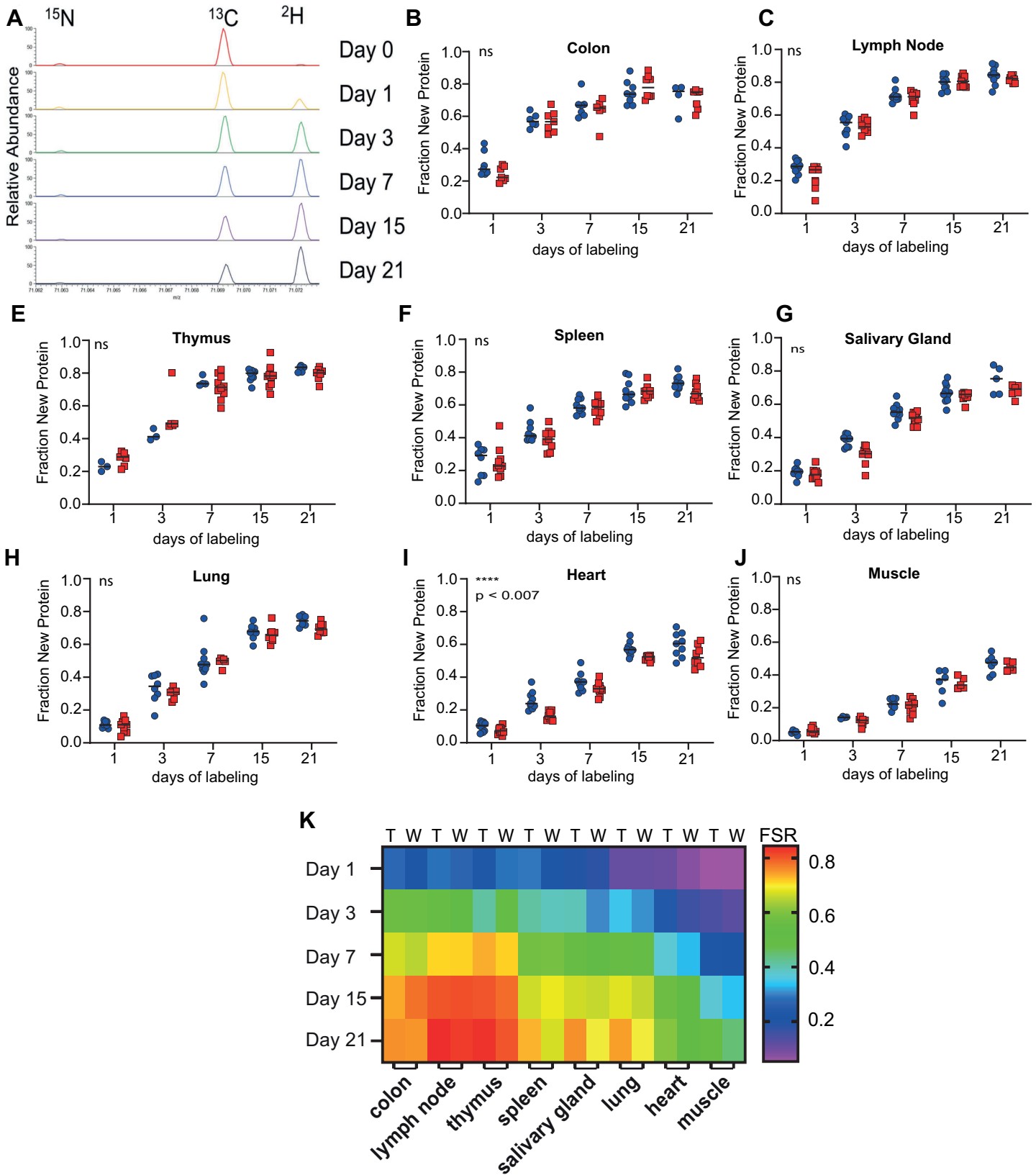

**Fig 3. STING FSR and model.** A) Representative spectra from STING peptide LILPGLQAR showing increase in 2H proline immonium ion peak. B–J) Time course of label incorporation (N = 5 per tissue) in WT (red squares) and TREX1$^{D18N}$ (blue circles) mice. Mean represented by black bar with statistical significance in FSR ($k$) reported as not significant, *ns*, or with denoted p-value. K) Heatmap of FSR calculated from model fit of data TREX1D18N (T) or WT (W) data in (B–J).

**Table 1. Table of STING protein flux: FSR and HALF-life.**

| Tissue | Gender | WT FSR (k) (Days⁻¹) | WT Half-life (Days) | TREX1^D18N/D18N FSR (k) (Days⁻¹) | TREX1^D18N/D18N Half-life (Days) |
|---|---|---|---|---|---|
| Colon | Male | 0.177 | 3.9 | 0.233 | 3.0 |
| | Female | 0.139 | 5.0 | 0.127 | 5.5 |
| Lymph Node | Male | 0.176 | 3.9 | 0.227 | 3.1 |
| | Female | 0.185 | 3.8 | 0.166 | 4.2 |
| Thymus | Male | 0.158 | 4.4 | 0.107 | 6.5 |
| | Female | 0.181 | 3.8 | 0.160 | 4.3 |
| Spleen | Male | 0.092 | 7.5 | 0.119 | 5.8 |
| | Female | 0.095 | 7.3 | 0.094 | 7.4 |
| Salivary Gland | Male | 0.078 | 8.9 | 0.098 | 7.1 |
| | Female | 0.083 | 8.4 | 0.100 | 6.9 |
| Lung | Male | 0.731 | 9.5 | 0.086 | 8.1 |
| | Female | 0.077 | 9.0 | 0.086 | 8.1 |
| Heart | Male | 0.045 | 15.4 | 0.055 | 12.7 |
| | Female | 0.046 | 15.1 | 0.056 | 12.3 |
| Muscle | Male | 0.027 | 25.4 | 0.030 | 23.4 |
| | Female | 0.031 | 22.1 | 0.032 | 21.9 |

Finally, to understand how STING's tissue specific protein turnover might play a role in drug pharmacology, we applied the measurements of STING's tissue specific FSR to model the recovery of STING protein following complete degradation resulting from exposure to a potent STING agonist. Here, we assumed complete loss of STING as reflected in the cellular and *in vivo* tumor model shown in Fig 1 and modeled the recovery of STING over time (Fig 5). In the lymph node, where the half-life of STING protein turnover is approximately 4 days, the full recovery of STING is expected to take ~15 days. The overall mouse FSR data and model presented here enables a relative prediction of STING protein levels in different tissues that can be applied to model pharmacology in different *in vivo* systems of STING dependent disease.

## Discussion

Protein turnover and fractional synthesis rate are dynamic properties of proteins that support drug disposition and pharmacology of disease relevant proteins. With STING-directed therapies in clinical trials, understanding STING's relative abundance and turnover is critically important to achieving target engagement at the most appropriate dose and frequency. To quantify STING's protein turnover in a disease-relevant model, we measured the fractional synthesis rates of STING in WT and Trex1^D18N mice. Two key observations were that STING's FSR is tissue specific and the FSR follows a similar trend in WT and chronically inflamed Trex1^D18N animals. Furthermore, in models of inflammation, higher levels of inflammatory gene transcription occur in tissues with the highest increase in STING protein abundance and with the slowest STING protein turnover. While our studies in mice are a starting point to scale STING protein turnover in human disease, its anticipated turnover rates in human will be slower due to the metabolic differences between rodents and humans [58].

The dynamic regulation of the proteome is a balance of degradation and resynthesis rates. In eukaryotes, degradation is primarily carried out by two major pathways: the

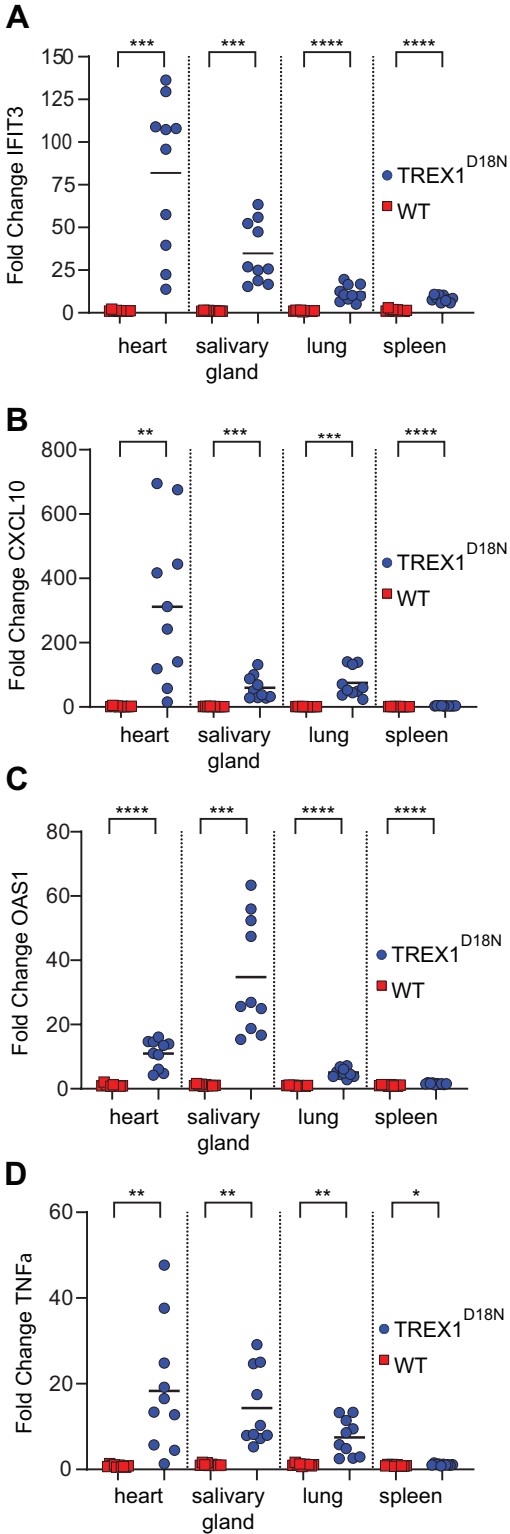

**Fig. 4. Quantification of tissue specific inflammatory gene transcription.** Fold change of gene expression derived from qPCR of interferon stimulatory genes (A) IFIT3, (B) CXCL10, (C) OAS1 and the inflammatory cytokine (D) TNFα from heart, salivary gland, lung, and spleen in WT (N = 10) and Trex1$^{D18N}$ mice (N = 10). Statistical analysis represents unpaired t-test significance of p < 0.0001 (****), 0.0001 > p < 0.001 (***), 0.001 < p < 0.01 (**), and p = 0.0324 (*).

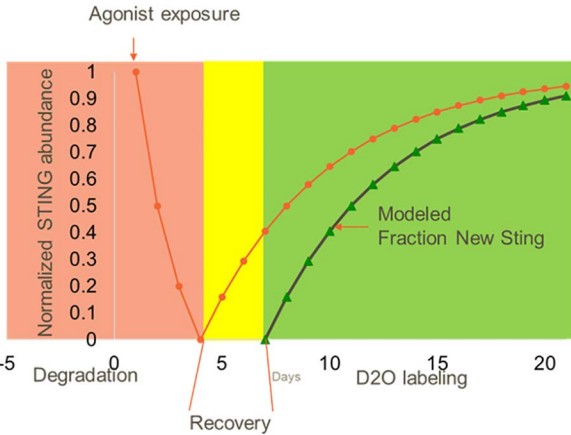

**Fig 5. Modeled STING recovery following STING agonist mediated protein clearance in lymph node.** Modeled STING recovery following agonist mediated protein clearance and subsequent metabolic labeling to measure STING resynthesis rate. STING protein is degraded following agonist activation (red area), followed by a recovery period where new STING synthesis occurs (yellow are). After a short recovery period, labeling occurs where resynthesis of STING is measured (green area). The pool size of STING (orange circles) drops due to activation and recovers following clearance of compound. Labeling during the recovery period, during which time the STING protein pool is replenished (green triangles) would enable assessment of the compound mediated effects on new synthesis following compound driven protein clearance.

ubiquitin-proteosome and lysosomal degradation. Under different cellular states, STING can be degraded by both. The most well characterized mechanisms of STING degradation describe the endolysosomal degradation of STING as a consequence of its innate immune stimulation [1,59,60]. In this regard, stimulation of STING follows a path of agonist ligand binding followed by conformational changes and ER-to-Golgi trafficking where STING interacts with TBK1 to phosphorylate and activate the transcription factors IRF3 and NFkB. What's striking about this mechanism is that ubiquitination of STING by multiple reported E3 RING-type ubiquitin ligases, including TOLLIP, RNF5, RNF144 and LUBAC, are thought to facilitate its regulated migration in vesicles to endolysosomes where it is degraded rather than targeted to the proteosome for degradation; the canonical endpoint for degradation of ubiquitinated proteins [51,53,61–63]. In the uninflamed state, the steady-state mechanism of STING protein degradation is much less well understood. One report demonstrated that the degradation of STING in human airway epithelial cells is regulated by the E3 ubiquitin ligase TRIM29 and is dependent on the 26S proteosome [64,65].

To understand the dynamics of STING protein in a whole body setting, we employed a metabolic labeling approach with deuterium oxide and analyzed the net effect of protein degradation and resynthesis using fraction synthesis rates. Newly synthesized STING was metabolically labeled over a time course following the administration of deuterium oxide in mice. Deuterium oxide was used as a universal stable isotope labeling reagent as it is convenient, can be consumed orally, is nontoxic, and rapidly equilibrates with total body water [66]. The deuterium from labeled water is incorporated into newly synthesized free amino acids and the incorporated deuterium does not back exchange with bulk water. Labelling of newly made amino acids occurs through the reactions of intermediary metabolism and the labeled new amino acids rapidly reach equilibrium with body water enrichment in an hour or two [67,68].

STING half-life varied between tissues ranging from ~4 days to ~23 days. The half-life reflects the steady state for STING abundance in the tissue where new STING synthesis and removal of existing STING are balanced. It is important to note that this method measures

bulk tissue dynamics of STING and cannot discern cell-type specific turnover rates nor cellular turnover events like immune cell trafficking to and from tissues during the time course of the experiment. A previous study of STING protein half-life in cycloheximide treated mouse embryonic fibroblasts reported a STING protein half-life of 12 hours [62]. This is dramatically shorter than the smallest half-life we report in colon and lymph node of 3 days and may be due to the limitations of using protein synthesis inhibitors in isolated cellular models for longer protein degradation rates [69]. Another study of STING protein half-life in primary human B-cells, NK cells, and monocytes reported STING protein half-lives of 347 hours, 720 hours, and 554 hours, respectively [55]. Given these differences and knowledge that tissue specific protein half-life can be context dependent, we set out to measure protein turnover in relevant pharmacological systems where targeted therapies may require a whole organism quantification of protein turnover. We find that even within the mouse, every tissue appears to have a distinct protein turnover for STING and underscores the need to measure STING protein turnover in target tissues to accurately predict drug-disposition in complex models of tumors or inflammatory disease.

From our measurements of mouse STING fractional resynthesis rates *in vivo*, we show that STING's bulk tissue turnover is overall similar in WT and inflamed Trex1$^{D18N}$ mice. This was somewhat surprising given that activation of STING is linked to its chronic activation, and presumably, its higher degree of degradation. There was a statistically significant difference in STING protein FSR in heart where the Trex1 dependent pathology is most severe and the causal decline in animal mortality [8,70,71]. Perhaps one explanation that more significant differences in FSR were not observed between WT and Trex1$^{D18N}$ animals may be due to the use of animals at 2 months of age where the onset for cGAS and STING dependent mortality and disease-related phenotypes begins in the Trex1$^{D18N}$ mouse [9,11,72]. Future studies to measure FSR for longer periods of disease duration will more clearly correlate the implications of heightened STING signaling and turnover in the Trex1$^{D18N}$ pathophysiology. We speculate that as disease-like phenotypes increase a greater influx of STING expressing immune cell infiltrate the heart from the periphery, diluting the pool of STING's cardiac muscle FSR and further deviating from the WT cohort.

The baseline abundance of STING was higher in Trex1$^{D18N}$ mice suggesting that increased immune cell infiltration and STING protein expression may be a bona fide hallmark of STING dependent inflammation. The amplitude of increased STING abundance in different tissues followed a similar trend with the amplitude of inflammatory gene expression. Specifically, the increase in transcription of IFIT3, CXCL10, OAS1, and TNFα was highest in heart followed by salivary gland and lung with the lowest response in spleen. This correlates well with the rank order of tissues that demonstrate the highest increase in STING protein levels in Trex1$^{D18N}$ compared to WT animals. Paradoxically, the levels of inflammatory gene expression were not directly correlated to the tissues with the highest total abundance of STING. For example, the most abundant STING protein levels in both WT and Trex1$^{D18N}$ are observed in spleen, yet the spleens showed basal levels of inflammatory gene transcription with the least significant differences between WT and Trex1$^{D18N}$. In terms of the fractional synthesis rate, the rank order of STING protein turnover appeared inversely proportional to the inflammatory gene transcription. The heart showed a lower STING FSR (longer half-life), yet it had the highest expression of IFIT3, CXCL10, OAS1, and TNFα compared to the spleen which has a STING FSR with a nearly 2-fold higher FSR. Perhaps a longer residence time of activated STING elicits a cellular program the generates the strongest inflammatory response. Given these observations, we propose that human disease tissues with the highest increase in STING levels may drive the highest inflammation and that tissues with the slowest STING protein turnover will also be more inflamed.

It is not uncommon for proteins to demonstrate different turnover rates in different tissues, cell types and under different environmental stressors [42,58]. For STING, the fractional synthesis rate was observed within a range of several days, measured in colon and thymus, to several weeks measured in skeletal muscle. Therefore, it is critically important to consider which tissue(s) and/or cell types are essential for STING's desired treatment endpoint. Given STING's role in driving chronic innate immune inflammation, it is a sensible hypothesis that a key driving force of STING-dependent inflammation is derived from circulating and/or tissue resident monocytes and macrophages where STING's expression is the highest and most biologically aligned with disease pathobiology. Here, it is likely that the circulating pool of monocytes and macrophage are represented by the splenic tissue derived STING turnover of 7.5 days. In affected organs, it is unclear whether tissue-resident monocytes and macrophages maintain their circulating FSR or adopt an FSR that is consistent with the overall tissue environment.

For our determination of STING FSR *in vivo*, we also considered the impact of STING's FSR in the context of a STING agonist treatment therapies. Here, we start to unravel the dynamics of the receptor and propose a model for calculating a meaningful dose frequency based on STING protein levels in specific tissues. As shown in Fig 1, it is possible that pharmacological activation will lead to partial or complete loss of STING. In this scenario, a dose frequency that meets the appropriate fractional resynthesis of STING must be considered before administering subsequent doses to drive the appropriate pharmacology. Consider the differences presented here for targeting STING in the lymph node versus the lung. In mouse lymph node, it will take ~8 days to fully recover from a complete loss of STING whereas in mouse lung it takes ~18 days. Scaling this to humans where the general proteome FSR is ~ 1.5 fold slower [58], these numbers scale to 12 and 27 days, respectively. Therefore, an agonist dosing regimen that elicits a complete response in lung after the first dose may require a one month dose holiday before a second administration. This may be underappreciated when measuring a systemic cytokine response as a marker for tissue specific target engagement since STING protein levels in blood may predominant the measured systemic response. Furthermore, relative consideration of tissue-specific FSR can have important implications for modeling safe dosing frequencies as a consideration to mitigate potential for excessive systemic cytokine secretion which is a common consideration among immune modulating drugs.

Understanding tissue specific STING half-life will help guide a more coordinated understanding for developing successful targeted interventions. The tissue specificity of STING half-life reflects a larger trend for proteins within tissues to have tissue dependent, distinct kinetic properties allowing for the generation of kinetic virtual biopsies [21,39]. Given the relationship between protein half-life and fractional synthesis rates, we can use the information presented here to predict how long a tissue will take to recover and have a new population of STING available for recurring dose interventions targeting STING. Knowledge of the paired compound distribution and target dynamics after interventions that cause activation and clearance will enable best practices and improved study design for drug development campaigns with a more precise targeting of drugs to the most appropriate site of action.

## Supporting information

**S1 Fig. Generation and characterization of Trex1[D18N].** (A) Illustration of targeting vector to generate the D18N mutation in the mouse Trex1 gene resulting in the mouse strain *C57BL/6NCrl*-Trex1tm1.1(D18N)Geno/Gsk. (B) Survival curves for WT (open circles) and Trex1[D18N] mice. (C) Tissue cytokine profiles for IP-10, MCP-1, and TNFα reported in WT (N = 10) and Trex1[D81N] (N = 10) animals. (E) Blinded histopathology score of heart, kidney, lung, salivary gland, and spleen from WT (white bars) and Trex1[D18N] (black bars). (EPS)

**S2 Fig. MRM methods of STING surrogate peptides.**
(TIF)

**S3 Fig. Replot of spleen ISG data on smaller y-axis scale.**
(EPS)

**S1 File. STING FSR raw data.** Spreadsheets containing raw data values for STING protein abundance, kinetics of fractional STING synthesis, and fold changes in RNA expression.
(XLSX)

## Acknowledgements

We thank Dr. Kate Fitzgerald and Dr. Fiachra Humphries for additional characterizations of the Trex1$^{D18N}$ mice previously published in (12). We also thank Dr. Julia Holter for input on statistics.

## Author contributions

**Conceptualization:** Thomas E. Angel, G. Scott Pesiridis.

**Formal analysis:** Thomas E. Angel, G. Scott Pesiridis.

**Investigation:** Thomas E. Angel, Zhuo Chen, Ahmed Moghieb, Sze-Ling Ng, Allison M. Beal, Carol Capriotti, Leonard Azzarano, Debra Comroe, Michael Adam, Patrick Moore, Bao Hoang, Kelly Blough, Joanne Kuziw.

**Methodology:** Thomas E. Angel.

**Supervision:** Thomas E. Angel, Sze-Ling Ng, Allison M. Beal, Joshi M. Ramanjulu, G Scott Pesiridis.

**Writing – original draft:** Thomas E. Angel, G Scott Pesiridis.

**Writing – review & editing:** Thomas E. Angel, G Scott Pesiridis.

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
