## [Decision Letter · Decision Letter 0]

12 Jun 2024

PONE-D-24-15873Implications of tissue specificSTING protein flux and abundance on the development of therapeutics.PLOS ONE

Dear Dr. Pesiridis,

Thank you for submitting your manuscript to PLOS ONE. After careful consideration, we feel that it has merit but does not fully meet PLOS ONE’s publication criteria as it currently stands. Therefore, we invite you to submit a revised version of the manuscript that addresses the points raised during the review process. Specifically, both reviewers have provided a well thought-out critic of the manuscript and have commented on background and foundational aspects of the study that need to be addressed as appropriate.

We look forward to receiving your revised manuscript.

Kind regards,

Jon M. Jacobs, Ph.D.

Academic Editor

PLOS ONE

-  Improved Sensitivity for Protein Turnover Quantification by Monitoring Immonium Ion Isotopologue Abundance - https://doi.org/10.1021/acs.analchem.9b01329.

(Among others)

In your revision ensure you cite all your sources (including your own works), and quote or rephrase any duplicated text outside the methods section. Further consideration is dependent on these concerns being addressed.

“All authors listed are current or former employees of GSK with shares in GSK plc.”

5. We note that your Data Availability Statement is currently as follows: [All relevant data are within the manuscript and its Supporting Information files.]

6. Please amend either the title on the online submission form (via Edit Submission) or the title in the manuscript so that they are identical.

7. We note that S1 Fig 2 in your submission contain copyrighted images. All PLOS content is published under the Creative Commons Attribution License (CC BY 4.0), which means that the manuscript, images, and Supporting Information files will be freely available online, and any third party is permitted to access, download, copy, distribute, and use these materials in any way, even commercially, with proper attribution. For more information, see our copyright guidelines: http://journals.plos.org/plosone/s/licenses-and-copyright.

1. You may seek permission from the original copyright holder of S1 Fig 2 to publish the content specifically under the CC BY 4.0 license.

8. We notice that your supplementary figures are included in the manuscript file. Please remove them and upload them with the file type 'Supporting Information'. Please ensure that each Supporting Information file has a legend listed in the manuscript after the references list.

Reviewers' comments:

Reviewer's Responses to Questions

**Comments to the Author**

1. Is the manuscript technically sound, and do the data support the conclusions?

Reviewer #1: No

Reviewer #2: Yes

2. Has the statistical analysis been performed appropriately and rigorously? 

Reviewer #1: Yes

Reviewer #2: Yes

3. Have the authors made all data underlying the findings in their manuscript fully available?

Reviewer #1: Yes

Reviewer #2: No

4. Is the manuscript presented in an intelligible fashion and written in standard English?

Reviewer #1: Yes

Reviewer #2: Yes

5. Review Comments to the Author

Reviewer #1: In this manuscript, Thomas Angel et al. use 2H2O to measure protein fractional synthesis rates (FSR) of STING across different tissues in vivo. The authors reached a conclusion that the FSR of STING varies from 0.027 in muscle to 0.731 in lung per day, and the half-life of STING varies from 3.8 to 25.4 days. Even unexpectedly, the half-life of STING in WT mice and STING-driven auto-immune TREX1D18N/D18N mice are largely comparable. Though the authors used cutting-edge techniques in the experiments, the conclusion is still not fully convincing, mainly due to very limited data provided and one single approach used.

1. The authors show the half-life of activated STING is 2 hours in Fig 1a, and this is consistent with most previous studies. The authors should use this widely-used system to compare with 2H2O-based techniques used in this study to verify the latter approach indeed works.

2. The authors also need more supportive data to verify their approach. For example, comparing their 2H2O-based techniques with standard CHX chase experiments to further validate their study. CHX chase experiments can at least be done in vitro using some culturable tissue or cells from WT and TREX1D18N/D18N mice.

3. The authors conclude the half-life of STING is at least 3.8 days, which is quite longer than the observed static STING half-life from CHX chase experiments in previous literature. Method-wise, mice were injected with 2H2O once followed by enriched drinking water (8% 2H2O) for the remainder of the experiment. Is this approach sufficient to make any amino acids in the mice incorporated with 2H2O during the whole experiment? If only a fraction of amino acids in the cell is incorporated with 2H2O and the unlabeled amino acids, for example, recycled from digested protein by the proteasome, can still be used in synthesizing STING, then the techniques used cannot detect the actual synthesis rate of STING, resulting in lower than the actual rate. This needs to be thoroughly addressed.

Reviewer #2: In this manuscript from researchers at GlaxoSmithKline, the authors document the protein turn-over of STING in wildtype (WT) C57Bl/6 mice and those carrying the Acardi-Goutiéres syndrome (AGS)-associated mutant isoform of Trex1 (D18N homozygous). The authors demonstrate that STING protein half-life is tissue specific and similar between WT and AGS mice, with the shortest half-life in colon and lymph node and longest in skeletal muscle. Overall, the experiments are well executed, and this study constitutes important but rather specialist information that will be useful to those researchers working on STING drug development.

Major Comments

1) The authors need to explicitly state where they obtained the Trex1 [D18N/D18N] model, as the details are not clear in the methods. Was this generated by the authors, did they receive it from a laboratory? What specific model is it? The original Trex1[D18N/D18N] model was in the 129S6/SvEvTac mouse background, not C57Bl/6. Did the authors obtain the mice already in the C57Bl/6 background, or did they do the backcrossed themselves. Really, much more detail is needed here to ensure future reproducibility and establish the provenance of the mice.

2) This is a rather limited manuscript in scope, and of VERY specialist interest. To move beyond specialist interest for drug development, the authors should provide correlation data on interferon stimulated gene (ISG) regulation down-stream of STING relative to protein turn-over and abundance in the tissues tested and between WT and AGS mice. This could be done using a basket of ISGs and a qPCR panel, or if budget allows via RNA-seq. This information would greatly increase the impact of the current study and interest for a broader range of researchers and in particular immunologists.

Minor Comments

1) Despite the misuse of human gene and protein (ALL CAPS) nomenclature by the original creators of the Trex1[D18N/D18N] mouse – the authors throughout the manuscript should use the correct gene and protein nomenclature for the mouse Trex1, which is capitalized first letter and then lowercase.

2) In the Author contributions please clarify what “…in-life TREX [D18N/D18N]…” means?

6. PLOS authors have the option to publish the peer review history of their article (what does this mean? ). If published, this will include your full peer review and any attached files.

**Do you want your identity to be public for this peer review?** For information about this choice, including consent withdrawal, please see our Privacy Policy .

Reviewer #1: No

Reviewer #2: No

---

## [Author Response · Author response to Decision Letter 1]

20 Dec 2024

Major edits and formatting are complete to meet the style requirements as proposed in the revised version of the manuscript and figures. In particular, changes were made to provide a separate title page with suggested formats, align Headings and Subheadings, figure references, placement of figure legends, font and citation styles, tables, figures and file naming standards. Please let me know if I’ve missed something.

- Improved Sensitivity for Protein Turnover Quantification by Monitoring Immonium Ion Isotopologue Abundance - https://doi.org/10.1021/acs.analchem.9b01329.

(Among others)

In your revision ensure you cite all your sources (including your own works), and quote or rephrase any duplicated text outside the methods section. Further consideration is dependent on these concerns being addressed.

The introduction was edited to rephrase text with some minor occurrence of overlapping text from the lead author’s previous publication cited above.

All funding for the studies was provided by GSK. There are no grant numbers to reference and have made more explicit statement in the ‘Funding Information’ section.

“All authors listed are current or former employees of GSK with shares in GSK plc.”

Please use the revised Competing Interests statement as suggested; “All authors listed are current or former employees of GSK who have owned shares in the past or currently hold shares in GSK plc. This does not alter our adherence to PLOS ONE policies on sharing data and materials.”

5. We note that your Data Availability Statement is currently as follows: [All relevant data are within the manuscript and its Supporting Information files.]

If there are ethical or legal restrictions on sharing a de-identified data set, please explain them in detail (e.g., data contain potentially sensitive information, data are owned by a third-party organization, etc.) and who has imposed them (e.g., an ethics committee). Please also provide contact information for a data access committee, ethics committee, or other institutional body to which data requests may be sent. If data are owned by a third party, please indicate how others may request data access.’

Yes, I confirm that all relevant data are within the manuscript and its Supporting Information files.

6. Please amend either the title on the online submission form (via Edit Submission) or the title in the manuscript so that they are identical.

The titles now agree.

7. We note that S1 Fig 2 in your submission contain copyrighted images. All PLOS content is published under the Creative Commons Attribution License (CC BY 4.0), which means that the manuscript, images, and Supporting Information files will be freely available online, and any third party is permitted to access, download, copy, distribute, and use these materials in any way, even commercially, with proper attribution. For more information, see our copyright guidelines: http://journals.plos.org/plosone/s/licenses-and-copyright.

I confirm that the aforementioned figure and has been removed.

1. You may seek permission from the original copyright holder of S1 Fig 2 to publish the content specifically under the CC BY 4.0 license.

8. We notice that your supplementary figures are included in the manuscript file. Please remove them and upload them with the file type 'Supporting Information'. Please ensure that each Supporting Information file has a legend listed in the manuscript after the references list.

Reviewers' comments:

Reviewer's Responses to Questions

Comments to the Author

1. Is the manuscript technically sound, and do the data support the conclusions?

Reviewer #1: No

Reviewer #2: Yes

2. Has the statistical analysis been performed appropriately and rigorously?

Reviewer #1: Yes

Reviewer #2: Yes

3. Have the authors made all data underlying the findings in their manuscript fully available?

Reviewer #1: Yes

Reviewer #2: No

4. Is the manuscript presented in an intelligible fashion and written in standard English?

Reviewer #1: Yes

Reviewer #2: Yes

5. Review Comments to the Author

Reviewer #1: In this manuscript, Thomas Angel et al. use 2H2O to measure protein fractional synthesis rates (FSR) of STING across different tissues in vivo. The authors reached a conclusion that the FSR of STING varies from 0.027 in muscle to 0.731 in lung per day, and the half-life of STING varies from 3.8 to 25.4 days. Even unexpectedly, the half-life of STING in WT mice and STING-driven auto-immune TREX1D18N/D18N mice are largely comparable. Though the authors used cutting-edge techniques in the experiments, the conclusion is still not fully convincing, mainly due to very limited data provided and one single approach used.

1. The authors show the half-life of activated STING is 2 hours in Fig 1a, and this is consistent with most previous studies. The authors should use this widely-used system to compare with 2H2O-based techniques used in this study to verify the latter approach indeed works.

The loss of STING upon stimulation in cell systems and in whole tumors shown in Fig 1 is a consequence of pathway stimulation and not a steady state turnover of the protein under sterile or unstimulated conditions. As noted, the point made in Fig 1 is in agreement with published reports (Gonugunta VK, Cell reports. 2017) and only characterizes the complete degradation of STING rather than the turnover which considers both resynthesis and degradation. We have added comments to the discussion to compare different methods to measure protein half-life along with limitations of classical methods that employ protein synthesis inhibitors in vitro. We also cite whole cell proteomics research in primary human cells that estimate the protein half-life to be on the order of hundreds of hours depending on the cell type (> 500 hours in primary human monocytes) using the approach published by Mathieson T, et al Systematic analysis of protein turnover in primary cells. Nat Commun. 2018;9(1):689.

The 2H2O approach used in this paper is a well-established and recognized method for the quantitative assessment of protein flux, both loss of protein and resynthesis, in vivo. This includes methods validated in rodents, monkey and man.

1) Shankaran, M., King, C. L., Angel, T. E., Holmes, W. E., Li, K. W., Colangelo, M., ... & Hellerstein, M. K. (2016). Circulating protein synthesis rates reveal skeletal muscle proteome dynamics. The Journal of clinical investigation, 126(1), 288-302.

2) Gasier, H. G., Fluckey, J. D., & Previs, S. F. (2010). The application of 2 H 2 O to measure skeletal muscle protein synthesis. Nutrition & metabolism, 7, 1-8.,

3) Miller, B. F., Reid, J. J., Price, J. C., Lin, H. J. L., Atherton, P. J., & Smith, K. (2020). CORP: The use of deuterated water for the measurement of protein synthesis. Journal of Applied Physiology, 128 (5), 1163-1176.

4) Holwerda, A. M., Atherton, P. J., Smith, K., Wilkinson, D. J., Phillips, S. M., & van Loon, L. J. (2024). Assessing muscle protein synthesis rates in vivo in humans: the deuterated water (2H2O) method. The Journal of Nutrition.

5) Holmes, W. E., Angel, T. E., Li, K. W., & Hellerstein, M. K. (2015). Dynamic proteomics: in vivo proteome-wide measurement of protein kinetics using metabolic labeling. Methods in enzymology, 561, 219-276.

6) Angel, T. E., Naylor, B. C., Price, J. C., Evans, C., & Szapacs, M. (2019). Improved sensitivity for protein turnover quantification by monitoring Immonium ion Isotopologue abundance. Analytical chemistry, 91(15), 9732-9740.

7) Fu, X., Deja, S., Fletcher, J. A., Anderson, N. N., Mizerska, M., Vale, G., ... & Burgess, S. C. (2021). Measurement of lipogenic flux by deuterium resolved mass spectrometry. Nature communications, 12(1), 3756.

2. The authors also need more supportive data to verify their approach. For example, comparing their 2H2O-based techniques with standard CHX chase experiments to further validate their study. CHX chase experiments can at least be done in vitro using some culturable tissue or cells from WT and TREX1D18N/D18N mice.

Processes that use transcription or translation inhibitors along with pulse-chase labeling approaches are limited to cellular systems and on a time-scale that is on the order of minutes and hours rather than days (which is required for accurate measures of STING protein in vivo). Furthermore, protein half-life measured by CHX pulse-chase approaches only capture the loss of protein, but fail to capture the tissue specific protein synthesis. Application of cycloheximide creates an artificial, non-biologically relevant, non equilibrium state for cells in culture and does not represent normal in vivo biology. Application of deuterium oxide at the levels used for metabolic labeling does not change the basal state of the organism allowing normal protein synthesis and degradation to occur over the time interval where protein dynamics are being measured. Please see Chan C, et al. Incompatibility of chemical protein synthesis inhibitors with accurate measurement of extended protein degradation rates. Pharmacol Res Perspect. 2017;5(5).

3. The authors conclude the half-life of STING is at least 3.8 days, which is quite longer than the observed static STING half-life from CHX chase experiments in previous literature. Method-wise, mice were injected with 2H2O once followed by enriched drinking water (8% 2H2O) for the remainder of the experiment. Is this approach sufficient to make any amino acids in the mice incorporated with 2H2O during the whole experiment? If only a fraction of amino acids in the cell is incorporated with 2H2O and the unlabeled amino acids, for example, recycled from digested protein by the proteasome, can still be used in synthesizing STING, then the techniques used cannot detect the actual synthesis rate of STING, resulting in lower than the actual rate. This needs to be thoroughly addressed.

Half-life of STING in tissues when measured in an intact animal is different than observed in immortalized cells or primary cells in culture. This is not unexpected as the metabolic rates

---

## [Decision Letter · Decision Letter 1]

29 Jan 2025

Implications of tissue specific STING protein flux and abundance on inflammation and the development of targeted therapeutics

PONE-D-24-15873R1

Dear Dr. Pesiridis,

We’re pleased to inform you that your manuscript has been judged scientifically suitable for publication and will be formally accepted for publication once it meets all outstanding technical requirements.

Kind regards,

Jon M. Jacobs, Ph.D.

Academic Editor

PLOS ONE

Additional Editor Comments (optional):

Reviewers' comments:

Reviewer's Responses to Questions

**Comments to the Author**

1. If the authors have adequately addressed your comments raised in a previous round of review and you feel that this manuscript is now acceptable for publication, you may indicate that here to bypass the “Comments to the Author” section, enter your conflict of interest statement in the “Confidential to Editor” section, and submit your "Accept" recommendation.

Reviewer #2: All comments have been addressed

2. Is the manuscript technically sound, and do the data support the conclusions?

Reviewer #2: Yes

3. Has the statistical analysis been performed appropriately and rigorously? 

Reviewer #2: Yes

4. Have the authors made all data underlying the findings in their manuscript fully available?

Reviewer #2: Yes

5. Is the manuscript presented in an intelligible fashion and written in standard English?

Reviewer #2: Yes

6. Review Comments to the Author

Reviewer #2: (No Response)

7. PLOS authors have the option to publish the peer review history of their article (what does this mean? ). If published, this will include your full peer review and any attached files.

**Do you want your identity to be public for this peer review?** For information about this choice, including consent withdrawal, please see our Privacy Policy .

Reviewer #2: No

---

## [Editor Report · Acceptance letter]

PONE-D-24-15873R1

PLOS ONE

Dear Dr. Pesiridis,

I'm pleased to inform you that your manuscript has been deemed suitable for publication in PLOS ONE. Congratulations! Your manuscript is now being handed over to our production team.

Kind regards,

on behalf of

Dr Jon M. Jacobs

Academic Editor

PLOS ONE